# Distributed Vibration Sensing Based on a Forward Transmission Polarization-Generated Carrier

**DOI:** 10.3390/s24165257

**Published:** 2024-08-14

**Authors:** Ming Chen, Xing Rao, Kuan Liu, Yuhang Wang, Shuqing Chen, Lin Xu, Rendong Xu, George Y. Chen, Yiping Wang

**Affiliations:** 1Key Laboratory of Optoelectronic Devices and Systems of Ministry of Education/Guangdong Province, College of Physics and Optoelectronic Engineering, Shenzhen University, Shenzhen 518060, China; 2210452010@email.szu.edu.cn (M.C.); 2250453012@email.szu.edu.cn (X.R.); 2110456067@email.szu.edu.cn (K.L.); 2100453052@email.szu.edu.cn (Y.W.); ypwang@szu.edu.cn (Y.W.); 2Shenzhen Key Laboratory of Photonic Devices and Sensing Systems for Internet of Things, Guangdong and Hong Kong Joint Research Centre for Optical Fibre Sensors, State Key Laboratory of Radio Frequency Heterogeneous Integration, Shenzhen University, Shenzhen 518060, China; 3Institute of Microscale Optoelectronics, Shenzhen University, Shenzhen, 518060, China; shuqingchen@szu.edu.cn; 4Jiangsu Ocean Technology and Equipment Innovation Center, Suzhou 215000, China; xulin@hmntech.com (L.X.); xurd@zju.edu.cn (R.X.); 5Ocean College, Zhejiang University, Zhoushan 316021, China

**Keywords:** distributed fiber-optic sensor, forward transmission, polarization-generated carrier, vibration sensor

## Abstract

For distributed fiber-optic sensors, slowly varying vibration signals down to 5 mHz are difficult to measure due to low signal-to-noise ratios. We propose and demonstrate a forward transmission-based distributed sensing system, combined with a polarization-generated carrier for detection bandwidth reduction, and cross-correlation for vibration positioning. By applying a higher-frequency carrier signal using a fast polarization controller, the initial phase of the known carrier frequency is monitored and analyzed to demodulate the vibration signal. Only the polarization carrier needs to be analyzed, not the arbitrary-frequency signal, which can lead to hardware issues (reduced detection bandwidth and less noise). The difference in arrival time between the two detection ends obtained through cross-correlation can determine the vibration position. Our experimental results demonstrate a sensitivity of 0.63 mrad/με and a limit of detection (LoD) of 355.6 pε/Hz^1/2^ at 60 Hz. A lock-in amplifier can be used on the fixed carrier to achieve a minimal LoD. The sensing distance can reach 131.5 km and the positioning accuracy is 725 m (root-mean-square error) while the spatial resolution is 105 m. The tested vibration frequency range is between 0.005 Hz and 160 Hz. A low frequency of 5 mHz for forward transmission-based distributed sensing is highly attractive for seismic monitoring applications.

## 1. Introduction

With the continuous advancement of smart cities, fiber-optic sensing technology is being ever more sought after and widely deployed in the field. Distributed fiber-optic vibration sensors leverage the high sensitivity and electromagnetic interference immunity of optical fibers to enable distributed monitoring of acoustic signals and vibrations in real time [1,2,3]. Low-frequency vibration monitoring is an important research area in industrial production, safety monitoring, seismic warning, and other fields [4,5]. Traditional low-frequency electronic vibration sensors suffer from various limitations, including the number of sensor nodes and multiplexing complexity [6]. In comparison, distributed fiber-optic vibration sensing systems can offer long-distance vibration mapping and safe deployment in combustive environments.

Conventionally, distributed vibration sensing is realized by sending an optical pulse along an optical fiber and collecting the backscattered light as a function of time. Alternatively, sweeping the laser wavelength and monitoring the beat frequencies is an option. Sensing systems based on Optical Time-Domain Reflectometry (OTDR) [7,8] and Optical Frequency-Domain Reflectometry (OFDR) [9,10] utilize a high-coherence laser source and weak Rayleigh backscatter; thus, the sensing range is inherently limited to tens of kilometers and several kilometers, respectively. Hybrid designs have also been explored to address weaknesses. For example, Chen et al. [11] proposed a distributed vibration sensor based on multi-pulse time-gated digital OFDR and reported a sensing length of 10 km with a spatial resolution of 10 m. Despite the excellent performance of OTDR and OFDR in analyzing short-distance ranges, they are not suited to long-range sensing due to the following reasons: (a) the weak backscattered optical signal and (b) the trade-off between sensing distance and spatial resolution/measurement rate. On the other hand, forward transmission-based [12,13,14,15] distributed fiber-optic vibration sensors are an emerging technology and have demonstrated much longer sensing distances (up to 200 km singe-span) through continuous-wave forward-propagating light and double-ended time-of-arrival-based vibration positioning. For instance, Zhao et al. [12] reported a distributed vibration sensor based on a dual Mach–Zehnder interferometer and a seven-core fiber, which demonstrated a sensing range of up to 38.5 km and a positioning error of 54.9 m. However, to push the upper limit of sensing distance even further, the signal-to-noise ratio must be improved.

To address the problem of SNR through bandwidth optimization, this paper proposes a solution in the form of a forward transmission polarization-initial-phase demodulation-based distributed fiber-optic vibration sensor. The SNR can be improved by reducing the detection bandwidth (hardware) since a large number of detection systems are shot noise dominated (white noise). This can be achieved by using a new method that only extracts the known carrier frequency followed by Fast Fourier Transform (FFT) analysis of its initial phase evolution instead of using the full detection bandwidth (higher noise) with FFT analysis. Carrier extraction can be performed by using an analytical discrete Fourier transform of a single frequency instead of a full FFT for faster computation (N compared to (N/2)log_2_N, or 2: log_2_N). Alternatively, a lock-in amplifier can isolate the frequency of interest to improve the SNR. Additionally, unlike OTDR and OFDR which exploit weak Rayleigh backscattered light, forward transmission type sensing typically employs continuous-wave transmitted light with optical power 3–4 orders of magnitude higher than that of Rayleigh backscatter, which provides a much longer single-span sensing range.

Compared with our previous polarization-type forward transmission-based distributed vibration sensor [13], the main differences in this work include the use of a completely different signal demodulation method (initial phase of a fixed-frequency polarization carrier instead of the polarization-rotation-induced power signal), and the polarization diversity analysis is replaced with a polarization analyzer.

## 2. Experimental Setup and Demodulation Principles

In this work, instead of analyzing an arbitrary signal frequency component, a higher-frequency polarization-generated carrier signal is introduced, which can be analyzed in the frequency domain using FFT on each data block (fast time, continuous sampling). Then, the initial phase of the carrier component is monitored (slow time, same index of each data block interval) to recover the vibration signal. The amplitude of the initial phase and frequency of the initial phase provide a reading of the vibration amplitude and frequency. The time-of-arrival difference of the same signal between the two detection ends provides the vibration position.

The polarization-generated carrier based on forward transmission is shown in Figure 1. The input light is split into two paths of equal power by a 3 dB fiber coupler. One beam propagates clockwise through the first circulator and transmits through the sensing fiber and then outputs from the second circulator. Similarly, the other direction of light propagates counterclockwise. The state of polarization of the two optical paths is modulated periodically after passing through the fast polarization controller and then converted into a power modulation by the inline polarizer. The optical signal is converted by the set of photodetectors into electrical voltage signals and then subsequently digitized by a USB oscilloscope connected to a computer. A piezoelectric transducer (PZT) with coiled fiber is used to simulate vibrations. Due to the different distances between the PZT and the two detectors, the sensing system can determine the position of the perturbation by the difference in time-of-arrival of the demodulated signal.

Unlike unwrapped phase measurements, the initial phase of a state of polarization (SOP) rotation-induced power signal is affected by the starting SOP angle, such that the PZT (first) alters the initial SOP azimuthal angle at the fast polarization controller (second), and thus the initial phase of the SOP-converted-power signal is a function of the PZT vibration frequency.

The periodic change in the SOP caused by the vibration signal applied by the PZT can be described by a Jones vector:(1)$$E=\left(\begin{array}{c} cos\left(\theta_{0}+Asin\;\left(2\pi ft\right)\right)\\ e^{i\delta }sin\left(\theta_{0}+Asin\;\left(2\pi ft\right)\right) \end{array}\right),$$
where δ represents the phase difference between the polarization states between the *x* and *y* directions, $\theta_{0}$ is the initial polarization angle of the input light, and *f* is the frequency of the polarization state change induced by external vibration signals.

The operation of the LiNbO_3_-based fast polarization controller is comparable to a cascade of five endlessly rotatable waveplates, allowing for control of the rotation angle of each waveplate to achieve any SOP. In sequence mode, the instrument can cycle through a sequence of SOPs at a chosen rate of up to more than 40 kHz. Switching between two predefined SOPs and cyclically switching between them allows for the generation of a polarization carrier. Assuming that the Jones matrix of the integrated system within the fast polarization controller can be regarded as *J*, then the initial SOP azimuthal angle to the FPC is influenced by the PZT vibration signal and the output SOP of the light passing through the FPC can be represented by the Jones vector:(2)$$E_{out}=J\times E=\left(\begin{array}{c} E_{x}\\ E_{y} \end{array}\right),$$

Assuming the angle between the output SOP and the polarizer axis is φ, the optical power of linear polarized light after passing through the polarizer can be calculated as follows:(3)$$P={\left|E_{x}\right|}^{2}\cdot {cos}^{2}\varphi +{\left|E_{y}\right|}^{2}\cdot {sin}^{2}\varphi ,$$

For demodulating the received voltage signal and extracting the initial phase, the flowchart is illustrated in Figure 2. Note that the SNR is higher than 20.6 dB in the low-frequency range, and generally, the higher the frequency, the higher the signal-to-noise ratio.

When vibration is applied to the sensing fiber with a length of *L*, the distance from the first and second circulators to the vibration position on the optical fiber are *x* and *L* − *x*, respectively. This leads to a time delay ∆*t* between the arrival time of the signals, which can be expressed as:(4)$$∆t=\frac{L-x}{\frac{c}{n}}-\frac{x}{\frac{c}{n}}=\frac{n}{c}\left(L-2x\right),$$

The vibration position *x* can be determined by the time delay ∆*t*:(5)$$x=\frac{L}{2}-\frac{c∆t}{2n},$$
where *c* is the propagation speed of light in vacuum, *n* is the effective index of the fiber core, and *L* is the length of the fiber.

## 3. Positioning Results

A linearly polarized coherent laser with an output power of 13 mW at a wavelength of 1550 nm and a linewidth of 50 kHz was used to probe the sensing system. The photodetectors have a bandwidth of 5 GHz. Figure 2a shows the voltage signal received by the oscilloscope, with a sampling rate of 1.98 MHz (sampling interval of 50 ms) and 99,210 sampling points in fast time per measurement. Figure 2b shows a magnified local region of Figure 2a, where each signal period contains 66 points due to the carrier frequency of the fast polarization controller being 30 kHz. FFT is performed on points 1 to 66, and the frequency with the maximum amplitude in Figure 2c was found, representing the vibration signal. A peak search algorithm was applied to select the exact time duration in order to ensure a constant initial phase in the case of no vibration. Then, the initial phase of this frequency was determined from the phase spectrum, which is shown in Figure 2d. By repeating this process for each measurement (sampling) in slow time, the initial phase of the carrier representing the signal frequency was obtained and plotted in Figure 2e. If a portable version of the sensing system is to be developed, only the known carrier frequency needs to be monitored for simplicity. It is possible to use a lock-in amplifier to beat with the known carrier frequency and shift the signal to a very low frequency, thus requiring a low-bandwidth detection system with much lower noise.

Due to the use of AC-coupling and filtering of the DC component, the zero-frequency signal is virtually non-existent and thus does not impact our detection results. The low-frequency detection goal is to decrease the vibration frequency signal as much as possible while ensuring a nominal signal-to-noise ratio of >20 dB. From the experimental results, the lowest frequency satisfying this condition is 5 mHz. Hence, for low-frequency demonstration, the vibration frequency of the PZT is set to 0.005 Hz. Using the polarization-initial-phase method, the measured signal frequency is in good agreement at 0.00485 Hz, as shown in Figure 3, with the discrepancy due to frequency quantization error. Figure 3 only shows the identified signal frequency, because cross-correlation is used, where only above-threshold signals that appear at both detection ends are plotted. The inset in Figure 3 shows the initial phase of the carrier at both ends.

To test the positioning ability of the system, the vibration is triggered at a distance of 69.986 km from the nominal end of the sensing fiber. Figure 4a displays the initial phase of the carrier obtained from polarization-initial-phase demodulation, while Figure 4b presents the cross-correlation between the demodulated signals from the two detection ends. Figure 4c shows the local magnification around the highest peak. The maximum peak corresponds to the time delay of −45.356 μs, which means the vibration position (x) by the cross-correlation algorithm yields a position of 70.288 km. Figure 4d shows the positioning result, which is affected by noise and ambient environment (e.g., temperature)-induced SOP changes. Compared to high-frequency signals (tens of kHz), low-frequency signals are more susceptible to phase noise in cross-correlation positioning, which affects positioning accuracy [14].

A linear fit was applied to the relationship between the applied strain and the corresponding phase shift, and the results are plotted in Figure 5. The minimum vibration amplitude is affected by phase noise, and the upper limit of vibration amplitude is limited to the available equipment, namely the maximum strain of the PZT used. The fiber-coiled PZT (pre-packaged) used in our experiments was calibrated by the manufacturer, and the relationship between applied voltage and measured PZT strain (radial) is known.

The sensitivity in our work is defined as the rate of change in the detected parameter (phase shift, rad) in response to changing strain. The derivation comes from the gradient of the linear fitting of the relationship between strain and phase shift. The strain values used in our work are that of the PZT since it is difficult to accurately determine the actual strain transfer to the bonded fiber. The phase response or sensitivity at 60 Hz is 6.3 × 10^−4^ rad/με. The standard deviation of phase noise can be determined from the standard deviation of noise-equivalent phase shifts in the absence of a vibration signal over a certain time period. Since the standard deviation of the phase noise is 0.0048 rad, the LoD can be calculated as follows (prior to bandwidth normalization):(6)$$LoD=3.3\times \frac{std\left(noise\right)}{sensitivty}=\frac{3.3\times 0.0048}{6.3\times 10^{-4}}\mu \varepsilon =25.1\;\mu \varepsilon ,$$

To investigate the frequency response of the sensing system, a series of measurements were conducted at the same position with low, medium, and high vibration frequencies. The corresponding sensitivity and LoD [16] were studied as a function of the vibration frequency. It can be seen from the experimental results in Figure 6 that the sensitivity increased and, consequently, the LoD decreased with increasing vibration frequency. This is because when the PZT experiences more abrupt movements with higher vibration frequencies, the adhesive layer has less time to deform, and thus transfer more strain from the PZT to the coiled optical fiber. When the vibration signal exceeds 500 Hz, the changes in sensitivity and LoD slow down, which is likely due to the frequency limitation of the PZT used.

To investigate the accuracy of vibration positioning, the vibration point was chosen at a distance of 70.59 km along the optical fiber, and the total length of the fiber was 131.5 km. In the experiment, the PZT was driven with a sinusoidal signal of 24 V amplitude and 100 Hz frequency. The sampling interval was set to 50 ms. Up to 50 consecutive measurements at the same vibration location were taken, and the root-mean-square (RMS) error of the measured result of x was also calculated to evaluate the positioning accuracy, which is given by
(7)$$X_{RMSE}=\sqrt{\sum_{i=1}^{k}\frac{{\left(x_{m,i}-\bar{x_{m}}\right)}^{2}}{k}},$$
where *k* is the number of measurements, i.e., *k* = 50, and $\bar{x_{m}}$ is the mean of the measured $x_{m}$. The RMSE of the positioning result was then estimated to be 725 m. The fiber length was verified using a commercial optical time-domain reflectometer (Anritsu MT9085, Atsugi, Japan) and cross-checked with the manufacturer’s datasheet. Note that the absolute difference is not used as the positioning accuracy since it can be calibrated based on the effective index of the fiber. Therefore, the root-mean-square error of the measurements from the peak position (725 m) was taken as the accuracy. There are two methods to improve the vibration positioning capability. One is to increase the polarization rotation frequency of the polarizer, which has high requirements for the hardware. The second is to use an interpolation algorithm to supplement the existing data for a finer spatial resolution. In addition, the spatial resolution is given by
(8)$$Z_{SPR}=\frac{c/n}{V_{SR}},$$
where the sampling rate ($V_{SR}$) of the oscilloscope is 1.98 MHz; thus, the spatial resolution based on the sampling interval is 105 m. Note that the phase-spectrum time delay method [17] in the frequency domain can be utilized in future work to considerably improve the spatial resolution.

The obtained experiment results expand the reported performance of distributed fiber-optic vibration/acoustic sensors in the areas of low frequency and long sensing distance (Table 1). Although the bandwidth advantage of the proposed sensing method does not contribute to any common sensing metric, it offers an alternative pathway for SNR optimization.

## 4. Conclusions

In conclusion, a new demodulation method for a distributed fiber-optic vibration sensing system based on forward transmission is introduced, which can potentially achieve long-distance sensing and ultra-low frequency vibration detection. The sensing mechanism involves the demodulation of the known-frequency SOP carrier rather than the unknown-frequency SOP signal. The initial phase of the carrier can be extracted using analytical discrete Fourier transform or a lock-in amplifier, and its time-dependent variation (containing vibration information) can be extracted using FFT analysis. This way, the detection bandwidth of the photoreceiver can be optimized based on the known frequency for noise reduction. The experimental results demonstrate that at low frequency (e.g., 0.005 Hz), the measurement sensitivity is 0.24 rad/mε and the LoD is 219.1 pε/Hz^1/2^. At medium frequency (60 Hz), the measurement sensitivity is 0.63 rad/mε and the LoD is 355.6 pε/Hz^1/2^. At high frequency (600 Hz), the measurement sensitivity is 4.1 rad/mε and the LoD is 905.5 pε/Hz^1/2^. The sensing range of the system can reach 131.5 km without optical amplifiers, with a positioning accuracy of 725 m and a spatial resolution of 105 m. In this experiment, vibrations in the range of 0.005 to 160 Hz were tested. The ability to detect 5 mHz vibration signals (SNR of 20.6 dB) allows for the monitoring of many seismic events in a wide range of applications from natural to man-made.

## Figures and Tables

**Figure 1 sensors-24-05257-f001:**
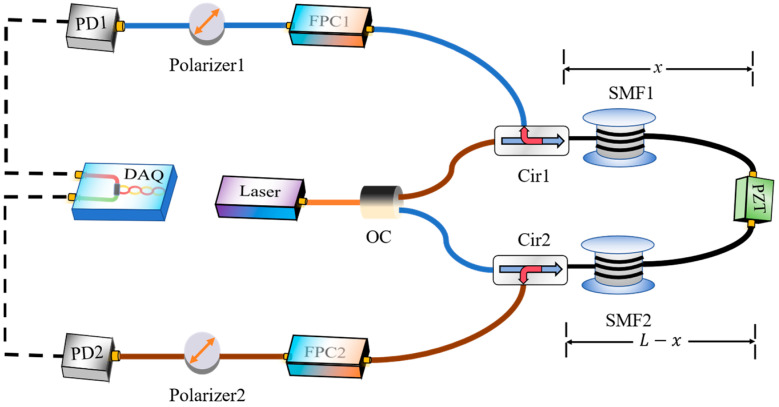
Schematic of the proposed distributed sensing system. OC: optical coupler, Cir: circulator, SMF: single-mode fiber, FPC: fast polarization controller, PD: photodetector, DAQ: data acquisition.

**Figure 2 sensors-24-05257-f002:**
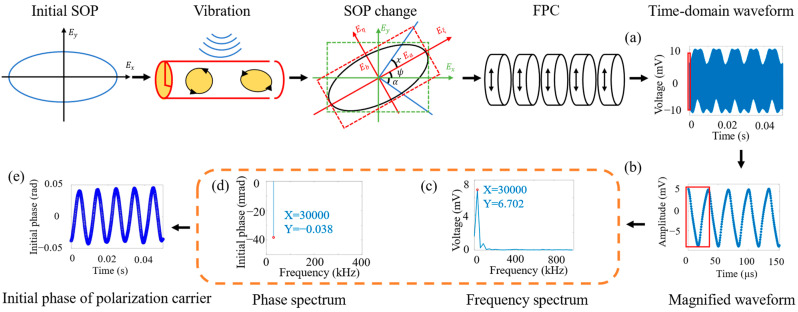
Flow chart of the polarization-initial-phase demodulation method. SOP: state of polarization, FPC: fast polarization controller. (**a**) PD voltage signal; (**b**) magnified local region of (**a**); (**c**) frequency with the maximum amplitude; (**d**) initial phase of nominal frequency; (**e**) initial phase signal. Note: The red box represents the fast Fourier transform results. Noise is not clearly visible due to the relatively high SNR.

**Figure 3 sensors-24-05257-f003:**
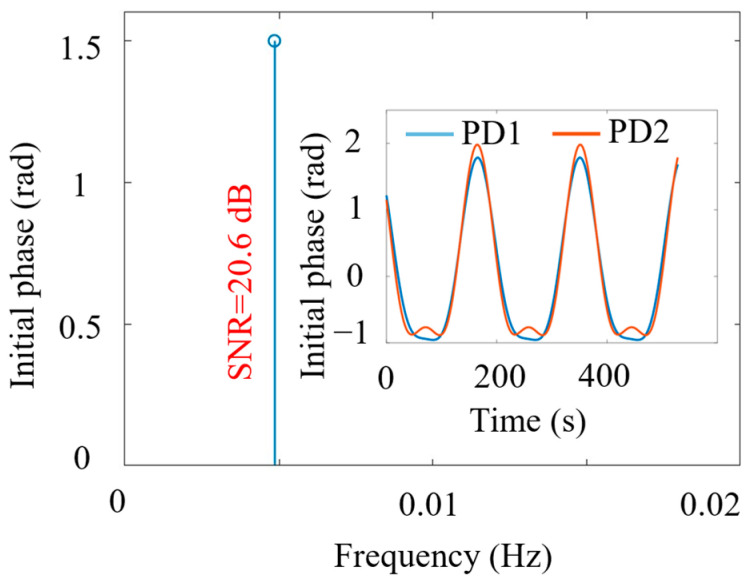
Vibration frequency observation from the time-domain waveform of the initial phase of the polarization carrier. Inset: initial phase signal. PZT was driven with 0.005 Hz.

**Figure 4 sensors-24-05257-f004:**
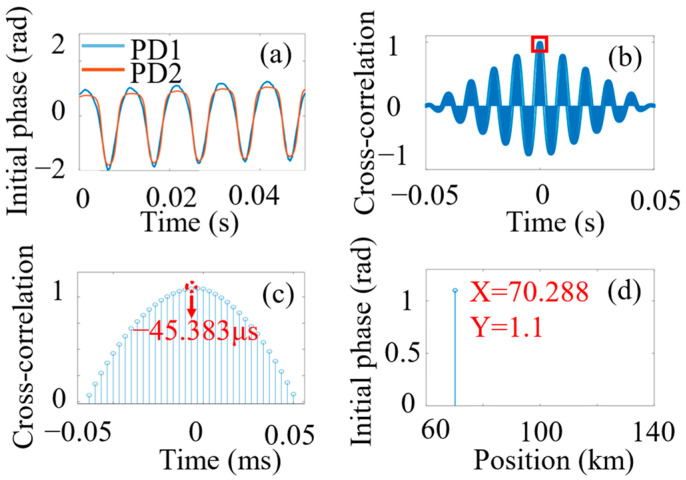
Vibration positioning process. (**a**) Measured vibration position; (**b**) initial phase signal; (**c**) cross-correlation results used for vibration positioning; (**d**) local magnification around the highest peak. PZT was driven with 100 Hz.

**Figure 5 sensors-24-05257-f005:**
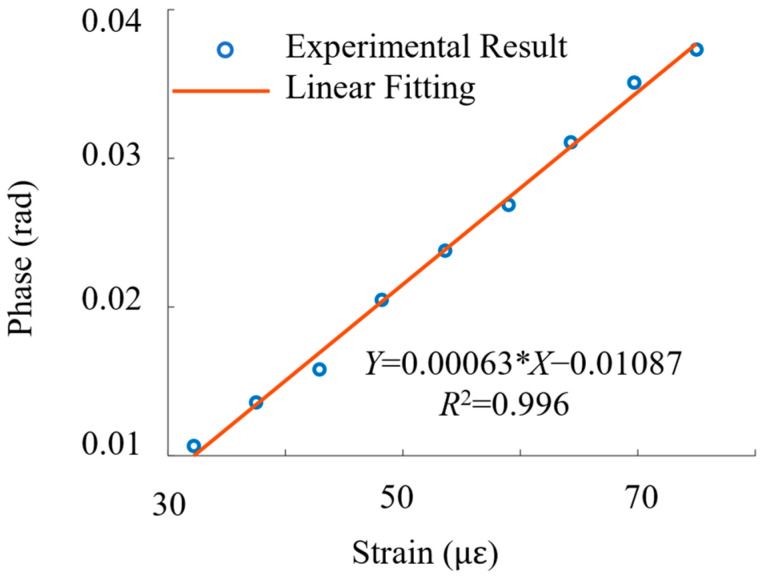
Linear fitting of the strain vs. phase shift relationship.

**Figure 6 sensors-24-05257-f006:**
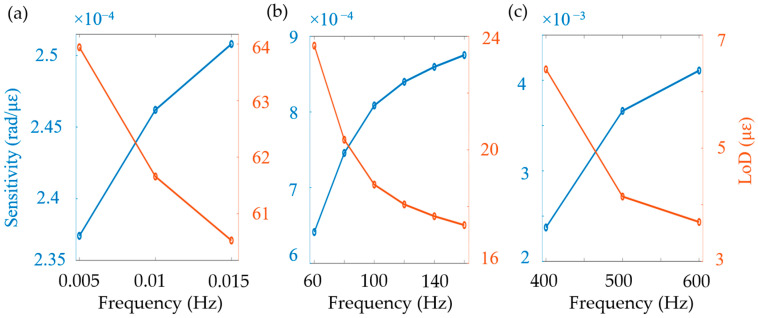
Sensitivity and limit of detection as a function of (**a**) low, (**b**) medium, and (**c**) high vibration frequencies.

**Table 1 sensors-24-05257-t001:** Comparison of key metrics in the literature.

Demodulation Scheme	Vibration Frequency Range	Sensitivity (PZT Coiled Fiber Length)	Sensing Distance	Positioning Accuracy	Reference
DAS based on a sub-chirped-pulse extraction algorithm	10–4900 Hz	80.7 pε/Hz^1/2^ @ 150 Hz (7.2 m)	0.92 km	0.284 m	[3]
Time-gated digital optical frequency-domain reflectometry	11–21 kHz	——	9.83 km	10 m	[11]
Dual-MZI interferometer using seven-core fiber	——	——	38.5 km	54.9 m	[12]
Polarization-demodulation based on WDM	20–100 Hz	1.6 rad/mε @ 100 Hz (20 m)	121.5 km	34 m	[13]
Initial phase of the polarization carrier	0.005–600 Hz	0.7 rad/mε @ 100 Hz (20 m)	131.5 km	725 m	This work

## Data Availability

Data underlying the results presented in this paper are not publicly available at this time but may be obtained from the authors upon reasonable request.

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
