# Peer review of "Distributed Vibration Sensing Based on a Forward Transmission Polarization-Generated Carrier"

_sensors, 2024, doi:10.3390/s24165257_

Round 1

Reviewer 1 Report

Comments and Suggestions for Authors

The work is devoted to the study of distributed sensing based on direct transmission combined with polarization carrier to reduce the detection bandwidth and cross-correlation for vibration positioning. The research has a right to exist, is a new approach to detect vibrations that can be caused by seismic activity, vehicular and airplane traffic, waves beating against the shore, mountain river noise, etc. etc. etc.

However, the article is not without a number of shortcomings:

(1) In the introduction it is advisable to present other vibration sensing devices (e.g., optical sensors, acoustic sensors, chemical sensors, etc.). It is necessary to compare the sensors (range, positioning accuracy), which will allow to identify the advantages and disadvantages of the proposed approach.

In conclusion, it is also necessary to present the advantages of the proposed vibration detection approach compared to known methods and approaches.

(2) In Section 3, it would be interesting to complement the studies by extending the frequency range (Fig. 5(b)) into a lower frequency region and a higher frequency region. Determine the output of the signal at the plateau. It would also be advisable to conduct studies as a function of vibration amplitude. It is clear that the minimum amplitudes will correspond to the sensitivity of the sensor. But the question of sensor operation in critical conditions at large vibration amplitudes remains.

Also of note are a number of technical errors that can be quickly corrected:

(1) It is necessary to increase the size of Fig. 1 so that the captions on the figures can be read

(2) line 112: formula (1) must be followed by a comma

(3) line 113: the line must start without indentation and with a small letter.

(4) line 125: formula (2) must be followed by a point

(5) line 129: formula (3) must be followed by a point

(6) line 143: formula (4) must be followed by a point.

(7) line 145: formula (5) must be followed by a comma.

(8) line 200: formula (6) must be followed by a point.

(9) line 219: formula (7) must be followed by a comma.

(10) line 231: formula (8) must be followed by a comma.

It is therefore recommended that additional information be added to the Introduction, Conclusion and Positioning Results

Comments on the Quality of English Language

Reviewer 2 Report

Comments and Suggestions for Authors

Comments on the Quality of English Language

The article has obvious grammar errors and needs careful editing.

Reviewer 3 Report

Comments and Suggestions for Authors

This paper introduces a new demodulation method for distributed optical fiber vibration sensing systems based on forward transmission of optical signals. By demodulating carrier signals with known frequencies, initial phase information is extracted. This method optimizes the detection bandwidth and may enable long-distance optical fiber sensing. It can also effectively detect extremely low frequency signals of 0.005Hz and has high sensitivity. However, there are still some questions, and it is recommended to receive the paper after revision.

1. How do the authors quantify the SNR value?

2. The author tested the detection of extremely low frequencies. During the Fourier transform, does the zero-frequency signal have an impact on the detection?

3. In Equation 6, how is the sensitivity defined and how is the standard deviation of phase noise obtained?

4. The results section can include more experimental data under different vibration frequencies and amplitudes to demonstrate the performance of the system under various conditions. For example, the system sensitivity and detection range at low and high frequencies can be compared.

Round 2

Reviewer 1 Report

Comments and Suggestions for Authors

The authors have made a change to the article. The article is a good option for perception. The article can be published in the present form.

Comments on the Quality of English Language

There are no notes to English in the article.